# Treatment of Periimplantitis with Electrolytic Cleaning versus Mechanical and Electrolytic Cleaning: 18-Month Results from a Randomized Controlled Clinical Trial

**DOI:** 10.3390/jcm10163475

**Published:** 2021-08-06

**Authors:** Markus Schlee, Hom-Lay Wang, Thomas Stumpf, Urs Brodbeck, Dieter Bosshardt, Florian Rathe

**Affiliations:** 1Private Practice Schlee und Rathe, 91301 Forchheim, Germany; thomas.stumpf@32schoenezaehne.de (T.S.); Florian.rathe@32schoenezaehne.de (F.R.); 2Department of Maxillofacial Surgery, Johann-Wolfgang-Goethe-University, 60590 Frankfurt am Main, Germany; 3Department of Periodontology, University of Michigan School of Dentistry, Ann Arbor, MI 48109, USA; homlay@umich.edu; 4Zahnmedizin Zürich Nord, 8051 Zürich, Switzerland; ursbrodbeck@bluewin.ch; 5Department of Periodontology, School of Dental Medicine, University of Bern, 3010 Bern, Switzerland; dieter.bosshardt@zmk.unibe.ch; 6Department of Prosthodontics, Danube University, 3500 Krems, Austria

**Keywords:** periimplantitis, electrolytic cleaning, air abrasive, augmentation, long term

## Abstract

**Aim of the study:** This RCT assesses patients’ 18-month clinical outcomes after the regenerative therapy of periimplantitis lesions using either an electrolytic method (EC) to remove biofilms or a combination of powder spray and an electrolytic method (PEC). **Materials and Methods:** Twenty-four patients (24 implants) suffering from periimplantitis were randomly treated by EC or PEC followed by augmentation and submerged healing. Probing pocket depth (PPD), Bleeding on Probing (BoP), suppuration, and standardized radiographs were assessed before surgery (T0), 6 months after augmentation (T1), and 6 (T2) and 12 (T3) months after the replacement of the restoration. **Results:** The mean PPD changed from 5.8 ± 1.6 mm (T0) to 3.1 ± 1.4 mm (T3). While BoP and suppuration at T0 were 100%, BoP decreased at T2 to 36.8% and at T3 to 35.3%. Suppuration was found to be at a level of 10.6% at T2 and 11.8% at T3. The radiologic bone level measured from the implant shoulder to the first visible bone to the implant contact was 4.9 ± 1.9 mm at mesial sites and 4.4 ± 2.2 mm at distal sites at T0 and 1.7 ± 1.7 mm and 1.5 ± 17 mm at T3. **Conclusions:** Significant radiographic bone fill and the improvement of clinical parameters were demonstrated 18 months after therapy.

## 1. Introduction

While dental implants have significantly changed the strategies used for the treatment of missing teeth, implant therapy is not without complications. Progressive bone loss around implants accompanied by inflammation has been described as “periimplantitis” [1,2]. The various different definitions of periimplantitis are causing there to be conflicting results in prevalent data. Therefore, workgroup 4 of the 2017 World Workshop on the Classification of Periodontal and Peri-Implant Diseases and Conditions initiated by numerous multinational dental societies created a new classification. Profuse bleeding and/or suppuration after careful probing and increased PPD and progressive bone loss after initial remodeling are the parameters for a positive diagnosis of periimplantitis in the case of patients under clinical control. If radiographic history of bone loss is not available, radiographic evidence of a bone level ≥ 3 mm and/or probing depth ≥ 6 mm in conjunction with profuse bleeding on probing can be classify a patient as having periimplantitis [3]. The primary etiology of periimplantitis remains a bacterial biofilm [4], with many possible contributing factors, such as iatrogenic issues (e.g., implant malposition), foreign body reactions, and debris particles. Hence, the clinician is faced with bone defects, inflamed soft and hard tissues, and bacterial biofilms contaminating implant surfaces. Therefore, the aim of a successful treatment of periimplantitis is the complete re-osseointegration of the implant surface or at least an elimination of the inflammation and stabilization of the clinical situation. Implant health can only be maintained if bone levels are stable over time and the soft tissue complex is not infected. The latter is difficult to manage over time if re-osseointegration is not achieved due to the possible reinfection of the implant surface [5], which may not be accessible to proper oral hygiene. Moreover, the bacterial biofilm and endotoxins need to be removed and the implant surface has to be put into a condition which allows a re-integration into the tissues [6,7]. This can be a significant challenge for the clinician to manage due to the defect morphology and the macro- and micro-morphology of the implant. Several mechanical methods for implant surface detoxification—most of them ablative—have been described in the literature, either performed as the only detoxification technique or in combination with other mechanical/chemical detoxification approaches. Bone fill has been proven radiologically, but animal studies have shown that most of the bone is not attached to the implant but rather separated by a more or less thick layer of connective tissue. Rates of reosseointegration of treated implant surfaces of between 39% and 46% have been reported in animal studies [8]. Reviews of literature following up the long-term results of different treatment modalities have demonstrated no proof of the superiority of any method. Furthermore, none of them have shown the ability to maintain infected implants over time with predictable results and low complication rates [9,10].

Initially introduced for removing stains from enamel surfaces, powder spray systems (PSSs) have been used in the treatment of periodontitis and periimplantitis [11,12]. PSS accelerate abrasive powders (sodium bicarbonate, sodium hydrocarbonate, erythritol, or amino acid glycine) using compressed air and water. In vitro and in vivo studies have demonstrated its ability to reduce biofilms [12,13]. Bone fill has been demonstrated in animal studies, but reosseointegration was limited to 39–46% and no additional benefit of air abrasive therapy compared to other treatment modalities was detected [8]. An in vitro study assessing the vitality and detectability of surviving bacteria after cleaning mature biofilms with a PSS showed disappointing results; nevertheless, this study confirmed PSSs’ limited cleaning efficacy [14]. Schlee et al. achieved complete reosseointegration after electrolytic cleaning in a preclinical study [15]. The clinical outcome of electrolytic versus electrolytic plus PSS cleaning was investigated by Schlee et al. in a prospective randomized clinical trial. The aims of this study were first to assess the efficacy of the electrolytic method (EC) in cleaning the contaminated implant surface and secondarily to evaluate if PSS provides any additional benefit. Bone gain visually and radiologically attached to the implant surface was assessed 6 months after treatment and submerged healing. Bone fill could be proven in both groups without reaching significant differences. In 50% of cases, a complete bone fill was observed [16]. The long-term clinical results of this cohort are very relevant for the evaluation of the technique. It has not yet been proven if these results can be maintained over time. Therefore, patients were reexamined clinically and radiologically 6 and 12 months after the replacement of restorative parts (this means 12 and 18 months after surgery). In contrast to regular augmentation procedures, the bony reconstruction of periimplantitis-related defects is compromised by inflammatory conditions in surrounding tissues [4]. Hence, the efficacy of the approach used to treat periimplantitis is primarily related to the successful elimination of infection and inflammation [17].

The aim of this study was to follow up the cases 18 months after treatment and 12 months after replacing the restoration to assess the clinical stability of the results.

## 2. Materials and Methods

### 2.1. Legal

This study was conducted according to the Helsinki declaration and complies with the consort checklist. The study was registered (BfArM DA/CA99, DIMDI 00010977) and approved by the “Ethik-Kommission der Bayerischen Landesärztekammer (BASEC-No. DE/EKBY10) with the registration code 17075.

### 2.2. Sample Size Calculation

Based on previous in vitro tests using a paired t-rest with a power of 90% and a level of significance of 5%, a sample size of 12 per group was used. The sample size calculation was conducted using G*Power 3.1.9.2 for Windows 10 (Heinrich Heine University of Düsseldorf, Düsseldorf, Germany).

### 2.3. Devices an Mode of Action

The mode of action of this electrolytic approach (EC) has been described before [16]. In brief, the infected parts of implants were loaded with a direct maximum negative current of 600 mA. A sodium formiate solution acting as an electrolyte is pumped by a device (GS1000, GalvoSurge Dental AG, Widnau, Switzerland) through a platinized ring acting as an anode and sprayed on the exposed and infected implant surface. Electrolysis produces hydrogen cations (H^+^) which penetrate the biofilm. Due to deoxidation, hydrogen bubbles emerge on the implant surface and lift the biofilm off the implant surface. This process leaves a clean implant surface with no visible, stainable, or breedable bacteria [14].

### 2.4. Patient and Sample Selection, Randomization

Twenty-four patients with periimplantitis (definition according to Berglundh et al.) [3] were enrolled in the study and allocated to the test group (EC) or control group (PEC). Sealed envelopes were used for randomization. If more than one implant was affected, one implant was chosen by dice roll. 

### 2.5. Inclusion Criteria

Patients older than 18 years who were able to understand and sign informed consent; who smoked less than 10 cigarettes per day; who had no uncontrolled periodontitis; who had a BoP < 20%, plaque index < 20%, and no allergy against the used drugs or materials; and who were not pregnant or nursing were suitable for enrollment in the study. Any patient with an infected implant, regardless of bone defect morphology, three-dimensional implant position, or inter-implant distance, was included in the study, in contrast to the protocols used in most of the literature.

### 2.6. Outcomes and Endpoints

BoP, PPD, suppuration, and radiologic bone levels at T0, T1, T2, and T3. 

### 2.7. Procedures and Measurements

Periodontal treatment was performed if necessary before patients were enrolled in the study. Suprastructures were removed 14 days before surgery (T00), and efforts were made to reduce inflammation (PSS cleaning (PerioFlow, Nozzle, Erythritol, EMS, Nyon, Switzerland) and rinsing with chlorhexidine (Chlorhexamed Forte 0.2%, GlaxoSmithKline Consumer Healthcare GmbH & Co. KG, Munich, Germany)). 

At baseline (T0), clinical photos were taken, and standard radiographs at right angles, suppuration, PPD, and BoP were assessed at 6 defined points (mesio-bucal, mid-buccal, disto-bucal, disto-lingual, mid-lingual, and mesio-lingual) using a periodontal probe with a 1 mm scale (PCPUNC 15, HuFridy, Chicago, IL, USA). The surgical approach used has been described before [16]. To summarize, a flap was raised after crestal incision, it was mobilized, granulomatous tissue was removed, and calculus was debrided if applicable. In the EC group, the implants were cleaned for 120 s electrolytically, as described before [16]. In the PEC group, powder spray (Airflow Plus powder, Airflow, EMS, Nyon, Switzerland) was applied to the infected implant surface according the manufacturer’s manual, followed by the treatment described for the EC group. Thereafter, the implants were augmented with autogenous bone harvested from the ramus area (Micross Safescraper, Zantomed, Duisburg, Germany), as well as deproteinized bovine bone mineral (DBBM) (Geistlich, Wohlhusen, Switzerland) in a 50:50 ratio. Sites were over-augmented up to 2 mm to ensure an adequate bone volume after shrinkage and healing. A collagen membrane (Bio-Gide, Geistlich, Wohlhusen, Switzerland) was used and, if necessary, tented up with Umbrella screws (Umbrella-Screw, Ustomed, Tuttlingen, Germany). The flap was sutured passively with 6-0 propylene monofilaments (Medipac, Kilis, Greece). Sutures were removed 2 weeks later. During the period of healing, the patients were supervised and exposures or infections were documented. Second-stage surgery was performed after 6 months (T1) and P-B was assessed. In exposed implants, P-B was assessed by bonesounding under local anesthesia. Furthermore, infections, BoP, and recessions were documented. Exposed implants were controlled according to the situation. Patients were instructed in personal hygiene measures and the exposed parts were cleaned by a hygienist with cleaning paste (Hawe Implant Paste, Kerr, Scafati, Italy) and rubber cups (Prophylaxe Gummikelche latexfrei, Alfred Becht GmbH, Offenburg, Germany). For all implants, restorative parts were replaced, photos were taken, and a standardized x-ray was performed in a right angle position. Sutures were removed after 14 days, if applicable [16].

PPD, BoP, suppuration, secretion, recessions, photos, and radiographs were recorded 6 months (T2) and 12 months (T3) later (12 and 18 months after baseline). Changes in those variables as well as radiologic bone levels were compared and statistically evaluated. For radiographic evaluation, DBS software (DBS Win, DentsplySirona, Bensheim, Germany) was used. As the diameter and length of the implants were known, the toll was calibrated to assess distances. To minimize examiner bias, they were not informed about the aim of the study and results were calibrated until they correlated adequately, as measured by Cohen’s Kappa (κ ≥ 0.6). In addition, the assessment data were tested for correlation (Spearman) and paired differences (Wilcoxon), and agreement between the two methods of clinical measurement was assessed (Bland-Altmann). 

Bone-level implants were judged to be completely osseointegrated if the bone level reached the platform. For implants with a polished neck, completely osseointegration was regarded as the bone level reaching the borderline of rough and polished [18,19]. All the surgical procedures and clinical assessments, such as PPD, recessions, and BoP, were performed by the first-named author in his private practice. Reducing surgery time is beneficial for the patient and the outcome, and adding a third blinded person to assess the clinical data would complicate the process. Therefore, we decided accept the potential bias that the surgeon assessing and performing the surgery was not blinded.

### 2.8. Statistics

The statistical analysis was conducted by an independent statistician. Quantitative values are presented as means and standard deviations, minimums and maximums, as well as quartiles. They were tested for normal distribution using the Shapiro-Wilk test, which is appropriate in the case of small samples. The Wilcoxon test was used to compare two related samples. McNemar’s test was used to test paired nominal data. A repeated measures ANOVA was used to compare the bone levels at different time points. For testing the accordance of the two assessors of radiologic bone levels, Spearman’s Rank correlation analysis, Wilcoxon matched pairs test, and Bland-Altman analysis were performed. The tests were two sided, with a significance level of 5%. An alpha adjustment for multiple testing was not applied, and the results were interpreted accordingly. Statistical calculations were carried out with SPSS Statistics 26 (SPSS Inc. an IBM Company, Chicago, IL, USA).

## 3. Results

Gender and age were distributed homogenously (12f/12m, age 57.13 y). Fourteen days after the removal of restorative parts and cleaning with PSS, all the sites were infected, BoP was positive, suppuration was drained from pockets, and all sites were probed deeper than 5 mm at baseline. PPD was 6.64 mm in EC and 7.02 mm in the PEC group at the median. Four patients (3 EC, 1 PEC) smoked <10 cigarettes per day. A total of 19 implants were exposed at suture removal, and 15 were exposed after 6 months. No implant was lost during the healing phase (Table 1).

The distribution of the implant morphologies, surfaces, designs, and surgical implications (implant position, soft tissue quality) varied. Statistically significant conclusions could not be drawn. A total of 19 implants were blasted and etched (different techniques), 2 were anodized, 2 were etched, and 1 had a HA coating. One implant had to be removed nine months after T2. Four implants had to be removed before T2 and one more implant before T3. All implants which had to be explanted were re-infected, and bone regeneration was incomplete. Further characteristics could be identified: exposures (7), periodontal history (5), previous augmentation (5), bad axis (3), placed too deep (1), and placed too high (1) (Table 2).

At T1, one patient did not show up for the appointment. As six implants (25%) had to be removed before T2 and 1 more implant were removed before T3 (total loss 29.2%) only 18 implants could be assessed. 

Figure 1 displays the consort flow chart of the study. 

The correlation between the assessments of the two investigators of the radiographs was almost perfect (R = 0.999, *p* < 0.001 (Spearman), *p* = 0.781 (Wilcoxon)). Figure 2 shows the Bland-Altman plot.

Figure 3 expresses the change in radiographic bone level at T0, T1, T2, and T3. Globally, the gain of bone at T1, T2, and T3 compared to at T0 was statistically significant (*p* < 0.001, ANOVA). Additionally, when comparing pairs (post hoc Bonferroni tests), significance was achieved in all pairs (*p* < 0.001).

Absolute numbers are expressed in Table 3. The negative values at T1 represent the over-augmentation, which was performed as described.

PPD was assessed at six points (m, mb, db, dL, L, mL). A significant reduction in probing depth was assessed at all probing points when T3 (mean 3.1 ± 1.4 mm) was compared to T0 (mean 5.8 ± 1.6 mm) (Wilcoxon test for pair differences, *p* < 0.001) (Figure 4). 

While BoP and suppuration at T0 were 100%, BoP was assessed at T2 to be 36.8% (seven sites) and at T3 BoP to be 35.3% (six sites). The values for pus were 10.6% (2 sites) at T2 and 11.8% (2 sites) at T3. Two sites were diagnosed as having a mild periimplantitis at T2. No statistically significant change was assessable for BoP (McNemar’s test, *p* = 0.275) and in the diagnosis of periimplantitis (Bowker’s symmetry test, *p* = 0.392) between T2 and T3. 

## 4. Discussion

A pre-requisite for the successful treatment of periimplantitis is the elimination of its provoking factor. This includes the complete debridement of the peri-implant defect, the thorough decontamination of the implant surface, and the removal of potential contributing factors, as well as protecting the wound environment during the healing phase (flap closure). Hence, the initial phase (T00) aimed to reduce inflammation. As PSS cleaning has been demonstrated limited benefit in the literature, we treated implants after removing restorative parts using a powder spray system. Nevertheless, positive BOP and pus were assessed in all of the implants at the surgery date. In our data, PSS failed to eliminate infection and inflammation after this 14-day period. Nevertheless, the investigator’s clinical feeling expressed a certain reduction in inflammation. More data should be collected to investigate if repeated pretreatment with PSS could reduce inflammation, which may be beneficial for later surgery. Initially, this study was designed as an RCT to assess differences in healing and efficacy after EC and PEC [16]. This demonstrated that there was no statistical difference between the groups, meaning that additional cleaning with PSS demonstrated no additional benefit. At T2 (12 months after surgery), 2 EC and 5 PEC implants had to be removed. In contrast to the majority of other studies, we did not exclude severe cases and other compromising factors. The bony reconstruction of periimplantitis-related defects is compromised by inflammatory conditions in the surrounding tissues [4] compared to augmentation procedures in conjunction with implant placement. This might be a causative factor for the high exposure rate found in or 6-month study and the rate of late implant loss seen in this study. It is not surprising that exposed implants achieved less bone gain and that all implants that had to be removed suffered exposure during healing. In further studies, attention must be turned to identifying the factors for reducing the exposure rate after flap surgery. The high rate of patients with a periodontal history is conspicuous. Whether the bacterial biofilm is the only causative factor or bone loss is caused by surgical, mechanical, or patient-related reasons and bacterial colonization occurs on the exposed surfaces secondarily [2] is still a matter of debate. This debate on etiology is not only an academic question but also influences the success rate of possible therapy due to possible different specimen susceptibility and uncorrectable surgical or mechanical obstacles. In any case, biofilms need to be removed to prevent the progression of disease or to treat periimplantitis successfully. Furthermore, the defect morphology has a major impact on the amount of bone regenerated on average and on the amount of complete reosseointegration [16]. Further studies will have to investigate the predictive and risk factors for treatment with EC, such as defect morphology, implant position, soft tissue properties, and implant hardware. These studies should aim to develop a clear treatment decision tree to help determine whether an implant should be removed or treated with EC. 

As additional PSS cleaning did not improve the outcome at the 6-month mark in regard to the amount of bone fill, we did not investigate the group differences in this follow-up study. As we achieved the same amount of bone fill and the same amount of still-exposed implant surface 6 months after treatment (T1) in both groups, no difference in the presented data was to be expected.

In the CE Mark approval study, all patients meeting the inclusion criteria were enrolled in this study despite uncorrectable factors, such as bad implant depth and axis, too thick a diameter, or severe defects that were difficult to augment. For this study, venous blood had to be drawn at several time points to prove that no cleaning solution or reaction products could be detected in the blood of the patient treated. For this reason, a waiting time of 30 min with no action had to be followed, which, among other things, extended the surgery time. This might explain the relatively high failure rate in the follow-up. If a good treatment outcome was the aim of a study, case selection should have been applied to optimize the cohort. This was not conducted, as already mentioned, and might be a risk for the interpretation of these data. Finally, we removed 29.2% of the treated implants according to our strict treatment protocol and the aim of our treatment—to achieve as much reintegration of the implants as possible. This may impact the statistical interpretation of the data.

Evaluation of bone levels based on radiologic findings is difficult. Casetta et al. compared clinical and radiologic measurements of bone levels and concluded that there was a statistically significant overestimation of the level of peri-implant marginal bone compared with surgical measurements [20]. Serino et al. confirmed the results and stated that there was an overestimation of 1–2 mm [18]. As all the radiologic measurements were performed by the same investigators for both groups and at all time points, the risk of bias during the assessment of differences in bone level should be negligible. A prerequisite for successful augmentation is providing immobile space and the stabilization of the blood clot by bone or a bone substitute. For this, Umbrella-Screws or 2 mm-high healing posts were used to tent up the augmented area if the defects were not self-contained. All of the cases were “over”-augmented up to 2 mm. This explains the “negative” bone loss at T1. As the bone is remodeled during healing, the bone height was reduced at T2 and T3 compared to at T1. These results outline the significant gain of radiologically assessable mineralized bone structure in contact with the implant surface. Of course, this is not a proof of re-osseointegration. Nevertheless, an animal study [15] and a histomorphometry of the explanted implants of this study proved the occurrence of successful re-osseointegration (this article has been submitted). Figure 5 expresses the clinical course of one of the failed cases. After electrolytic cleaning and augmentation, the flap was exposed during wound healing. Part of the bone graft was lost and an incomplete bone fill was achieved. Over time, the implant surface which was not covered by bone was recolonized by bacteria. After 11 months, the implants had to be removed because of infection, suppuration, and swelling. The patients agreed on the histologic evaluation of the implant and the surrounding bone. The sites were not augmented during the initial implant placement. This proves that the area where bone substitute could be found must be new bone achieved after treatment. The crestal part of the histology clearly demonstrated new bone attached to the implant. To the best of the author’s knowledge, the re-osseointegration of an implant treated for periimplantitis has never been described before in a published study. This happened despite an ongoing infection crestal to the investigated area.

For the determination of long-term results, changes in the clinical parameters PPD, BoP, and suppuration are relevant. In the World Workshop on the Classification of Periodontal and Peri-Implant Diseases and Conditions, implants were judged healthy if PPD was <6 mm and no suppuration or bleeding occurred [3]. Carcuac et al. [19] achieved an improvement in PPD and BoP when comparing different cleaning modalities combined with apical flap surgery from T0 at 6 months and 1 year (PPD 7.82 ± 1.52 mm, 5.11 ± 1.71 mm, and 5.24 ± 1.97 mm; BoP 100%, 52%, and 41.9%). Suppuration changed from 68.7% to 14.4% and 17.4%. 

In our study, the surviving implants showed a mean PPD of 3.1 ± 1.5 mm in all sites after 1 year (T2). While suppuration and BoP at T0 were 100%, both parameters were reduced at T2 and T3, with no statistical difference between the latter (McNemar’s test, *p* = 0.275) (BoP/T2: 36.8% (7 sites), suppuration/T2: 10.6% (2 sites) and BoP/T3 35.3% (6 sites), and suppuration/T3 11.8% (2 sites)). No statistically significant change was assessable for BoP (McNemar’s test, *p* = 0.275) and in the diagnosis of periimplantitis (Bowker’s symmetry test, *p* = 0.392) between T2 and T3. Only two sites (11.8%) were diagnosed as mild periimplantitis, and two (11.8%) were diagnosed as mucositis. Our surgical protocol aimed at bone regeneration and required a coronal advanced flap. Carcuac et al.’s approach aimed at pocket reduction using an apical repositioned flap usually causing a reduced pocket depth and thus reduced inflammation. Our outcome is better than that of the cited clinical study, although even the apical repositioning of the flap was avoided by us. Comparing our results using EC with those using other treatment modalities is difficult because the aims of those approaches differ [10]. Pisano et al. performed a review and meta-analysis to compare the laser treatment of Periimplantitis with conventional approaches and found no difference regarding PPD and CAL [21]. Heitz-Mayfield and Mombelli conducted a systematic review and stated that, due to the heterogeneity of the study designs, case definitions, and outcome variables, no meta-analysis was possible and no specific recommendation for a treatment modality could be made [22]. These results were confirmed by Koo et al., who stated that no data exist to favor any decontamination protocol or prove the superiority of one treatment protocol [23]. The implemented clinical studies aimed at a reduction in inflammation. We aimed to achieve the reintegration of the implants in the tissues and can assume with some probability that reosseointegration may occur. Future studies will have to investigate if submerged healing is beneficial compared to open healing. The latter would be easier to handle because restorative parts would not have to be removed. Future studies with bigger cohorts are recommended in order to develop a clear treatment tree based on predicting factors such as defect morphology, implant position, soft tissue properties, and implant hardware. Positive results might encourage the community to investigate larger case series or even, through translational medicine, transfer the results to orthopedic and trauma surgery, where medical devices made of metal also suffer in biofilm-induced infections.

## 5. Conclusions

The electrolytic cleaning of contaminated implants decontaminates the implant surface, which allows the re-integration of the implant in the surrounding tissues. Significant radiologic gains in bone and reductions in PPD, BoP, and suppuration were proven and remained stable over an 18-month period. 

## Figures and Tables

**Figure 1 jcm-10-03475-f001:**
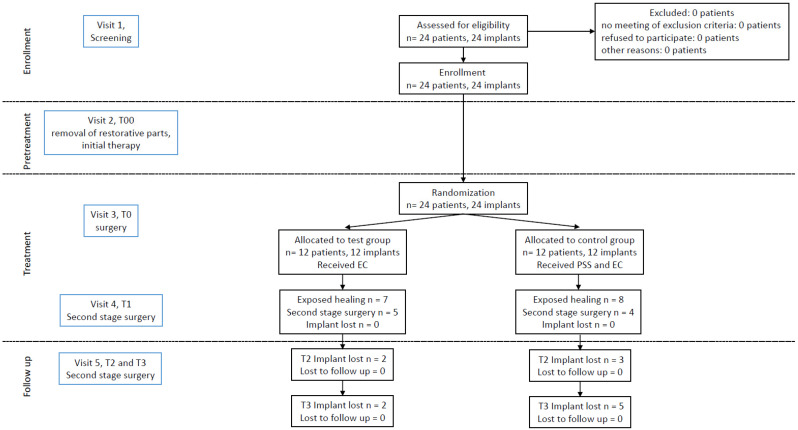
Consort flow chart of the study.

**Figure 2 jcm-10-03475-f002:**
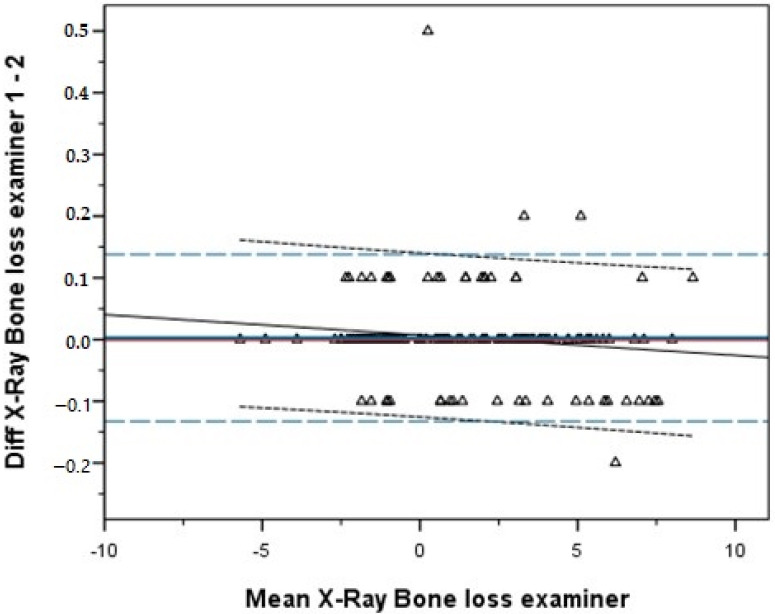
Bland-Altman plot.

**Figure 3 jcm-10-03475-f003:**
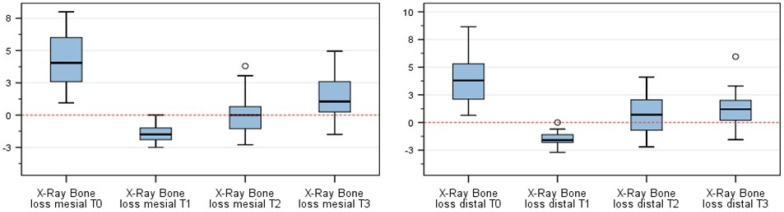
Bone level changes at the assessed time points. “°” = outlier.

**Figure 4 jcm-10-03475-f004:**
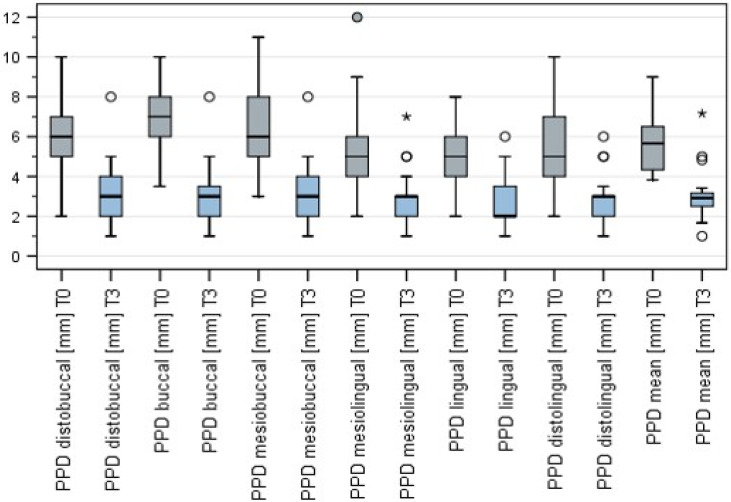
Changes in PPD at T0, T1, T2, and T3. “°” = outlier, “*” = extremum.

**Figure 5 jcm-10-03475-f005:**
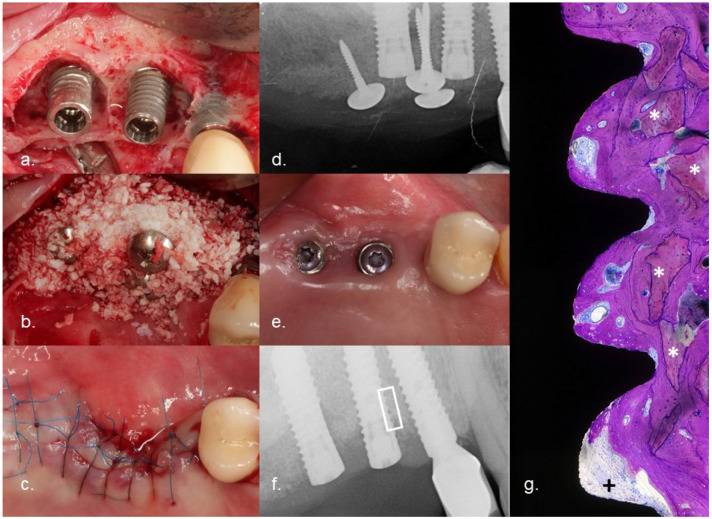
Clinical course of a failed case: (**a**) implants 15 and 16 with severe combined intra- and supra-osseous bone defects (RP Class 3); (**b**) augmentation after electrolytic cleaning with a combination of mineralized bovine bone mineral and autogenous bone using the Umbrella technique; (**c**) tensionless suturing after the crestal mobilization of the flap; (**d**) radiograph demonstrating the augmented volume; (**e**) clinical situation 6 months after the flap surgery, exposure of the flap caused some loss of augmented volume; (**f**) radiograph 11 months after surgery (the implant had to be removed because of suppuration); (**g**) histology with a field of interest in the augmented area pointing out regenerated bone and reosseointegration to the implant surface. * Particles of deproteinized bovine bone mineral, +pus.

**Table 1 jcm-10-03475-t001:** Overall patient data at T0.

		N/Mean Years/mm	Percentage
gender	female	12	50.00%
	male	12	50.00%
age	female	59.2 y	
	male	51.44 y	
jaw	maxilla EL	4	16.67%
	maxilla PEL	8	33.33%
	mandible EL	8	33.33%
	mandible PEL	4	16.67%
	maxilla total	12	50.00%
	mandible total	12	50.00%
smokers	EL	3	12.50%
	PEL	1	4.17%
BoP		(T00/T0) 24/24	100.00%
PUS		(T00/T0) 24/24	100.00%
PPD	EL	6.64 mm	
	PEL	7.02 mm	

**Table 2 jcm-10-03475-t002:** Explanted implants.

Patient	EC (Test)	PEC (Control)	Sex	Periodontal History	Bad Axis	Bad Diameter	Placed too Deep	Placed too High	Previous Bone Augmentation	Exposure
1	1	0	f	1	0	0	0	0	1	1
2	0	1	f	1	0	0	0	0	1	1
3	0	1	f	0	1	0	1	0	1	1
4	1	0	m	1	1	0	0	0	1	1
5	0	1	f	1	1	0	0	1	0	1
6	0	1	f	0	0	0	0	0	0	1
7	0	1	f	1	1	1	0	0	1	1
total	2	5	6 f, 1 m	5	3	1	1	1	5	7

f = female, m = male.

**Table 3 jcm-10-03475-t003:** Bone level changes from T0 to T3.

	N	Mean	Std.-Deviation	Minimum	Maximum	Perzentile		
						25	50 (Median)	75
X-ray Bone loss mesial T0	24	4.85	1.91	0.95	8	3.13	5.2	6.41
X-ray Bone loss mesial T1	23	−1.63	1.10	−4.9	0	−2	−1.5	−1
X-ray Bone loss mesial T2	24	0.69	1.93	−2.3	3.8	−0.88	0.6	2.76
X-ray Bone loss mesial T3	18	1.66	1.74	−1.5	4.95	0.19	133	3.3
X-ray Bone loss distal T0	24	4.38	2.16	0.65	8.5	2.81	4.5	5.55
X-ray Bone loss distal T1	23	−1.62	1.12	−5.7	0	−1.9	−1.6	−1
X-ray Bone loss distal T2	24	1.13	1.80	-2.2	4.1	−0.65	1.3	2.88
X-ray Bone loss distal T3	18	1.47	1.86	−1.55	5.95	0.15	1.33	2.68

## Data Availability

Data available on request due to restrictions—e.g., privacy or ethical.

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
