# Peer review of "Treatment of Periimplantitis with Electrolytic Cleaning versus Mechanical and Electrolytic Cleaning: 18-Month Results from a Randomized Controlled Clinical Trial"

_jcm, 2021, doi:10.3390/jcm10163475_

Round 1

Reviewer 1 Report

Dear Authors, the article is well organized and written. 
Please, add in the references this recent review: 

Pisano M, Amato A, Sammartino P, Iandolo A, Martina S, Caggiano M. Laser Therapy in the Treatment of Peri-Implantitis: State-of-the-Art, Literature Review and Meta-Analysis. Applied Sciences. 2021; 11(11):5290.

Author Response

Thank you for your input. The article was added to discussion

Reviewer 2 Report

Der authors,

Congratulations on your research project.

Thank you for submitting your paper to this journal.

The reviewer would like to suggest, in case the authors would consider to continue to research the topic in the future, to maybe consolidate the findings by implementing biofilm identification (PCR or any other technologies). 

Author Response

Thank you for the recommendation. We will consider to implement this recommendation in further studies. Those are planned for the therapy of infections in orthopedic surgery

Reviewer 3 Report

The present study describes an interesting approach in a field, where currently no decontamination method is scientificly described as superior over the other. Surely, this study represents an important effort in periimplantitis treatment. Nevertheless, important information have to be added and major revisions have to be applied.

  1. Abstract: If the authors want to keept he first sentence they should change “Background“ into „Aim oft he study“ for example.
  2. Introduction: Authors should reference the statements in the first sentences (line 41-49).
  3. Please revise the sentence written in line 87-89.
  4. The sentence in line 96-98 is formulated very unclear.
  5. Line 126 : Berglundh is written with an « h » on the end. In general revise for typos and grammar errors. I suggest a professional English proofreading.
  6. Power analysis was carried out on which primary outcome? What is primary and what is secondary?
  7. Formulate the null-hypothesis.
  8. How were radiographs standardized?
  9. Please describe in which setting patients underwent surgery. Private practice? More than one center?
  10. Line 156: Authors should write deproteinized bovine bone mineral (DBBM) (Bio-Oss, Geistlich etc...)
  11. Describe, if there was maintenance between augmentation and T1. Were plaque scores assessed ? please discuss.
  12. Please describe results seen in table/figure 3 more precise.
  13. Describe the percentage of failures, this should also be mentioned in the abstract and conclusion.
  14. Discuss the rational behind:„For tissue level implants, it is counted as 1mm below the platform 178 due to initial biological bone remodeling.“
  15. Describe the implant morphologies and surfaces included.
  16. Discuss the rational behind removing the prosthetics. Is there any advantage described by removing?
  17. Describe the morphologies found during surgery (in results) compare morphologies with failures and discuss the implication for augmentative procedures.
  18. „All the surgical procedures and clinical assessments such as PPD, recessions and 181 BoP were performed by the first-named author.“Mention and discuss a potential bias, as it was not blinded.
  19. Discuss the droputs and the statistical implications as 25% of the patients droped-out.
  20. „Further studies will have to investigate the risk factors for treatment with EC. These 279 studies should aim to develop a clear treatment decision tree whether an implant should 280 be removed or treated.“ Please be more clear in this statement.
  21. „The rate of implants which 293 had to be removed in the next 102/161cases/implants which were treated after this study 294 in our clinic was seven out of 161 implants.“ This statement does not rely on scientific data and should therefore be removed.
  22. In general discussion should be rearranged in order to gain a more homogenous “story”.
  23. Results should not only be compared to 1 further study only
  24. Outline if less failures were reached with the EC/PEC method compared to other decontamination methods. This should also be stated in the conclusion part.

Author Response

Thank you for your valuable comments, which we gladly implemented.

  1. Abstract: If the authors want to keept he first sentence they should change “Background“ into „Aim oft he study“ for example.

Done. Background was changed in Aim of the study.

  1. Introduction: Authors should reference the statements in the first sentences (line 41-49).

Two references added

  1. Please revise the sentence written in line 87-89.

Done. Now the sentence makes sense:

The clinical outcome of electrolytic versus electrolytic plus PSS cleaning was investigated by Schlee et al. in a prospective randomized clinical trial.

  1. The sentence in line 96-98 is formulated very unclear.

Improved into:

Therefore the patients were reexamined clinically and radiologically 6 and 12 months after replacement of restorative parts (this means 12 and 18 months after surgery).

  1. Line 126 : Berglundh is written with an « h » on the end. In general revise for typos and grammar errors. I suggest a professional English proofreading.

Changed in Berglundh and proofreading is organized

  1. Power analysis was carried out on which primary outcome? What is primary and what is secondary?

The original CE-mark study aimed on bacterial load before and after therapy. The null-hypothesies was: between the two groups (EC and EC + PSS) is no difference regarding bacterial load assessed by PCR test.

This was how the patients have been selected. In the article published first (6 month results) those two groups were followed up. The primary outcome in this article [1] was bone level changes, the secondary change of clinical parameters (PPD, BoP, suppuration).

The recent submitted article follows up the patient 18 months after surgery (12 months after load).

The primary outcome again is bone level changes, the secondary change of clinical parameters (PPD, BoP, suppuration)

  1. Formulate the null-hypothesis.

This study has no null-hypothesis because it was a follow up of the 6 month results which was a follow up of the CE mark study. In the original study the null-hypothesis was: between the two groups (EC and EC + PSS) is no difference regarding bacterial load assessed by PCR test.

  1. How were radiographs standardized?

To clarify this the following sentence was added:

As the diameter and length of the implants were known the toll was calibrated to assess distances.

  1. Please describe in which setting patients underwent surgery. Private practice? More than one center?

In line 182 the following was added:

in his private practice

  1. Line 156: Authors should write deproteinized bovine bone mineral (DBBM) (Bio-Oss, Geistlich etc...)

done

  1. Describe, if there was maintenance between augmentation and T1. Were plaque scores assessed ? please discuss.

The following sentences were added to line 165 to clarify:

Exposed implants were controlled as the situation required. Patients were instructed in personal hygiene measures and the exposed parts were cleaned by a hygienist with cleaning paste (Hawe Implant Paste, Kerr, Scafati, Italy) and rubber cups (Prophylaxe Gummikelche latexfrei, Alfred Becht GmbH, Offenburg, Germany).

  1. Please describe results seen in table/figure 3 more precise

Following sentences were added to clarify:

The negative values at T1 represent the over-augmentation which was performed as described [16] to ensure enough bone volume after healing. Even though implants had to be removed the median bone level at T2 and T3 improved significantly compared to T0.

  1. Describe the percentage of failures, this should also be mentioned in the abstract and conclusion.

The text was changed in:

At T1 one patient did not show up for the appointment. Because 6 implants (25%) had to be removed before at T2, only 23 and 1 more implant was removed before (total loss 29,2%) T3 only 18 implants could be assessed.

Figure 1 displays the consort flow chart of the study. 

  1. Discuss the rational behind:„For tissue level implants, it is counted as 1mm below the platform 178 due to initial biological bone remodeling.“

There is no rationale - the sentence in nonsense. Thank you for noticing. The sentence was removed.

  1. Describe the implant morphologies and surfaces included.

After table 1 the following sentence was added:

The distribution of the implant morphologies, surfaces, designs and surgical implications (implant position, soft tissue quality) varied. Statistical significant conclusions could not be drawn. 19 implants were blasted and etched (different techniques), two were anodized, 2 were etched and one had a HA coating.

  1. Discuss the rational behind removing the prosthetics. Is there any advantage described by removing?

This was described in the 6 month article which was cited several times. We thought it would be too iterative to repeat that. The rationale was to be able to cover the surgical site to achieve a submerged healing and to optimize success rate.

We added in discussion:

Future studies will have to investigate if a submerged healing is beneficial compared to open healing. The latter would be easier to handle because restorative parts would not have to be removed. 

  1. Describe the morphologies found during surgery (in results) compare morphologies with failures and discuss the implication for augmentative procedures.

This was described in the 6 month article which was cited several times. We thought it would be too iterative to repeat that. This article aims to follow up a longer time period. We strongly agree that defect morphology is one of the key factors to predict the amount of bone which can be regenerated and by this the success rate. The number of patients was too small to draw statistical relevant conclusions. For this we follow up a cohort of meanwhile more than 200 implants to be able to recommend a clear treatment tree.

We added in discussion:

Future studies with bigger cohorts are recommended to develop a clear treatment tree based on predicting factors like defect morphology, implant position, soft tissue properties and implant hardware.

  1. „All the surgical procedures and clinical assessments such as PPD, recessions and 181 BoP were performed by the first-named author.“Mention and discuss a potential bias, as it was not blinded.

We added the following in line 181:

Reducing surgery time is beneficial for the patient and outcome and implementing a third blinded person assessing the clinical data would complicate the process. So it was decided accept the potential bias that the surgeon assessing and performing the surgery was not blinded.

  1. Discuss the droputs and the statistical implications as 25% of the patients droped-out.

This is an important question which was addressed in the discussion widely. 25% respectively 29,2 % sounds very high. This needs to be discussed openly. On the other hand there were clear reasons for this numbers as discussed (no case selection, high surgery time due to study design,…). In the next cases we followed up now (almost 200 implants) the failure rate was drastically lower. We removed this hint, as you suggested. Of course we will come back with this numbers in the next article. Now we have to be clear to avoid a bias by putting this unnormal high numbers in the mind of the readers.  

We added to line 297:

Finally we removed 29,2 % of the treated implants according to our strict treatment protocol and our aim of the treatment – to achieve as much reintegration of the implants as possible. This is consequent but may impact the statistical interpretation of the data. 

  1. „Further studies will have to investigate the risk factors for treatment with EC. These 279 studies should aim to develop a clear treatment decision tree whether an implant should 280 be removed or treated.“ Please be more clear in this statement.

We changed to:

Further studies will have to investigate the predicting and the risk factors for treatment with EC such as defect morphology, implant position, soft tissue properties and implant hardware. These studies should aim to develop a clear treatment decision tree whether an implant should be removed or treated with EC

  1. „The rate of implants which 293 had to be removed in the next 102/161cases/implants which were treated after this study 294 in our clinic was seven out of 161 implants.“ This statement does not rely on scientific data and should therefore be removed.

Done, but this cohort is followed up according scientific principles and will be submitted for publication within this year.

  1. In general discussion should be rearranged in order to gain a more homogenous “story”.

done

  1. Results should not only be compared to 1 further study only
  2. Outline if less failures were reached with the EC/PEC method compared to other decontamination methods. This should also be stated in the conclusion part.

  1. and 24 are followed bay adding this text to discussion:

To compare our results using EC with other treatment modalities is difficult because the aim of those approaches differs [2]. Pisano et al. performed a review and meta-analysis to compare laser treatment of Periimplantitis with conventional approaches and fond no difference regarding PPD and CAL [3]. Heitz-Mayfield and Mombelli conducted a systematic review and stated that due to the heterogeneity of the study designs, case definitions and outcome variables no meta-analysis was possible and no specific recommendation for a treatment modality could be given [4]. This results were confirmed by Koo et al. who stated that no data exist to favor any decontamination protocol or prove the superiority of a treatment protocol [5]. The implemented clinical studies aimed on reduction of inflammation [4][4]. We aimed on reintegration of the implants in the tissues and can assume with some probability that the reosseointegration may occur.

References

  1. Schlee, M.; Rathe, F.; Brodbeck, U.; Ratka, C.; Weigl, P.; Zipprich, H. Treatment of Peri-implantitis-Electrolytic Cleaning Versus Mechanical and Electrolytic Cleaning-A Randomized Controlled Clinical Trial-Six-Month Results. Journal of clinical medicine 2019, 8, doi:10.3390/jcm8111909.
  2. Esposito, M.; Grusovin, M.G.; Worthington, H.V. Treatment of peri-implantitis: What interventions are effective? A Cochrane systematic review. Eur J Oral Implantol 2012, 5 Suppl, S21-41.
  3. Pisano, M.; Amato, A.; Sammartino, P.; Iandolo, A.; Martina, S.; Caggiano, M. Laser Therapy in the Treatment of Peri-Implantitis: State-of-the-Art, Literature Review and Meta-Analysis. Applied Sciences 2021, 11, 5290, doi:10.3390/app11115290.
  4. Heitz-Mayfield, L.J.A.; Mombelli, A. The therapy of peri-implantitis: a systematic review. Int. J. Oral Maxillofac. Implants 2014, 29 Suppl, 325–345, doi:10.11607/jomi.2014suppl.g5.3.
  5. Koo, K.-T.; Khoury, F.; Keeve, P.L.; Schwarz, F.; Ramanauskaite, A.; Sculean, A.; Romanos, G. Implant Surface Decontamination by Surgical Treatment of Periimplantitis: A Literature Review. Implant Dent. 2019, 28, 173–176, doi:10.1097/ID.0000000000000840.

Round 2

Reviewer 3 Report

Thank you for the adaptations. i suggest to accept the manuscript, but you should further apply minor revisions:

  1. Add the number of failed treatments in the abstracts results section. Despite it is not a main parameter, it should still be mentioned.
  2. Line 214.. a ")" is missing.